# A Study on the Strain-Softening Constitutive Model of Cementitious Sandstone

Zhongguang Sun [1,2,3,*], Kequan Wang [1,2,3,*], Yueming Kang [1,2,3] and Wanli Hu [3]

1   State Key Laboratory of Coal Mine Disaster Dynamics and Control, College of Resources and Environmental Science, Chongqing University, Chongqing 400044, China; kangyaoming@126.com
2   State Key Laboratory of The Gas Disaster Detecting, Preventing and Emergency Controlling, Chongqing 400037, China
3   China Coal Technology and Engineering Group Chongqing Research Institute, Chongqing 400039, China; hwl-1125@163.com
*   Correspondence: sunzhongguang126@126.com (Z.S.); kingcasey@163.com (K.W.)

**Abstract:** In order to further investigate the impacts of the water environment on the mechanical properties of rocks for an engineering project, taking the water-rich conditions in a coal mine as the engineering background, a series of tests were conducted, including the uniaxial compression test, the conventional triaxial compression test, and the constant axial pressure test on the cementitious sandstone. This was conducted along with the establishment of a multi-linear strain softening constitutive model. According to the tests, the following conclusions can be drawn. Firstly, as the water content increases, the weakening effect of water on the rock mass was obvious. Under various stress paths, the water weakened the rock body to various degrees. In other words, the weakening effect of water on the rock mass was either inhibited or promoted under different stress path conditions Secondly, under various stress paths, the turning point strength and strength variance rate of the rock mass' mechanical properties decreased linearly with the increase of water content. This further proves that water has a weakening effect on the rock mass, showing that the failure of the specimen changes from brittleness to ductility. Thirdly, the test sample demonstrated different types of damages including the tensile failure, transformation from tensile-shear composite failure to shear failure, and expansion failure under three stress path conditions. In addition, the unloading process demonstrated some dynamic failure characteristics. The research aims to provide some foundational insights for the scientific design and safe construction of the mine and other underground engineering, especially rock mass engineering in the multi-water environment.

**Keywords:** water content; cementitious sandstone; mechanical characteristics; constitutive model; failure mode





## 1. Introduction

As Qian Qihu, one of the winners of the State Preeminent Science and Technology Award of China, pointed out, China has been a major player in exploring the underground [1]. China also has the most rock related projects completed and in the process of construction [2]. The research on rock mechanics is indispensable for underground projects. Despite progress made in rock mechanics and engineering projects, a number of technical difficulties are encountered. Due to the rising needs of underground exploration and various technical difficulties, gaining an understanding of rock strains, rock strengths, and rock mechanical properties under various conditions has become a critical and urgent task [3,4].

Rock, as one of the anisotropic and heterogeneous natural materials, is featured with complex structures, which adds difficulties to underground engineering. Under most circumstances, such complex and unpredictable properties of rock mass are often impacted by the water environment. In recent years, underground engineering water disasters have

attracted great attention from the world. According to relevant statistical data, as high as 30% to 40% of dam damages and 80% of rock slope failures are related to impacts of water weakening on the rocks to various degrees [5–7].

Scholars domestically and abroad such as Price N.J. [8], Feda J. [9], Colback [10], Z.A. Erguler [11], Á. Török [12],Chen Ganglin [13], Liu Jian [14], Yongchuan Zhao [15], Louis Ngai Yuen Wong [16], etc. have invested substantial efforts in exploring the physical impacts of water on its mechanical properties. According to existing studies, the compressive strength and tensile strength of rock samples tend to decrease as the water content increases. The water also contributes to the degrading of friction coefficient, elastic modulus, Young's modulus, and the viscosity coefficient of the rock to various degrees. Professor Kang Hongpu [17] identified a linear relationship among the uniaxial compressive strength and elastic modulus of rock and the water content, along with the identification of a method using the water contents to represent different levels of rock damage. Zhu Zhende et al. [18] suggested a non-linear relationship between the uniaxial compressive strength of mud slate and water content along with a basic linear relationship between the elastic modulus and water content. Based on the elastic modulus, an evolution equation was developed to describe the water damage variable of rocks under the conditions of rock expansion, bulk density, and stress state.

Some significant contributions have been made by scholars domestically in terms of the water-rock physics of expansive rocks. Ji Ming et al. [19] studied the time effect of lime mudstone expansion in contact with water and proposed the concept of a time threshold for rock expansion stability. Based on the analogy between the humidity stress field and the moisture stress field in governing differential equations, Lu Aihong et al. [20] stimulated the moisture stress field indirectly through the simulation of the temperature stress field. Yu Shilian and Mao Xianbiao et al. [21] performed the numerical calculation of the coupling problem of swelling and in-situ stress of expansive rock tunnels. Tang Anchun et al. [22] established a humidity, stress, and rock rheological model to explore stress-damage coupling action by taking advantage of the moisture diffusion and rheological effects. Ren Song [23] deduced the roadway stability of the tunnel under the expansion effect of anhydrite surrounding rocks. Some scholars conducted a series of hydraulic and mechanical tests on the fractured rocks under the hydraulic pressure in various joints. They expounded the microstructure deterioration and macroscopic mechanical softening mechanism of the water-soaked rocks and proposed a variety of rock coupling models of seepage flow field and stress field [24–27].

In this study, a series of tests were conducted including the uniaxial compression test, the conventional triaxial compression test, and the constant axial pressure test on the cementitious sandstone. This was conducted along with the establishment of a multi-linear strain softening constitutive model, aiming to study the strain-softening constitutive model of cementitious sandstone and failure modes. The study provides some insights regarding the control of the surrounding rock in the water-rich environment of the Gucheng coal mine, other similar underground projects, and the numerical simulations of moisture expansion.

## 2. The Test Design of Cementitious Sandstone

The cementitious sandstone used in the experiment was selected from the #3 coal seam roof of the Gucheng Coal Mine located at Changzhi, Shanxi Province, China, with a buried depth of about 500 m and an average density of 2.44 g/cm$^3$. According to the X-ray diffraction pattern, the rock sample was mainly composed of plagioclase, quartz, and calcite, including a 12% of kaolinite mixed with chlorite, Mongolia, Yi/Mongolia, and Kaolinite (which is often easy to become soft and muddy when contacting with water). According to the recommendations of the International Society of Rock Mechanics on the property test of rock specimens (1979) and the requirements of Standards for Test Methods of engineering rock masses (GB/T 50266-2013), the rocks were processed into cylindrical standard specimens in a dimension of Φ50 mm × 50 mm with a height–diameter ratio of 2:1, as shown in Figure 1.

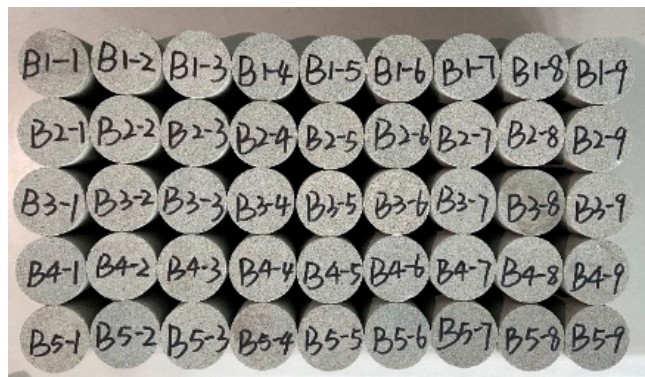

**Figure 1.** Standard specimen of cementitious sandstone.

### 2.1. The Measurement of Water Content

1.  Firstly, the specimens were heated in the constant temperature oven (model XMTD-8222) at 105 °C for 48 h. The initial water content in the specimens was zero. The specimens were removed from the oven and cooled down to room temperature (25 °C). The parameters of the specimens including the quantity m1, diameter D1, and height h1 were recorded.
2.  Secondly, under room temperature, specimens were soaked in the purified water (to avoid any impacts of PH on the rock strength 27) for 0 h, 0.75 h, 1.5 h, 6 h, 24 h, 48 h, and 30 days/720 h. The specimens were removed from the water. The parameters of the specimens including the quantity m2, diameter D2, and height h2 were recorded.

The drying process and soaking process were demonstrated in Figure 2.

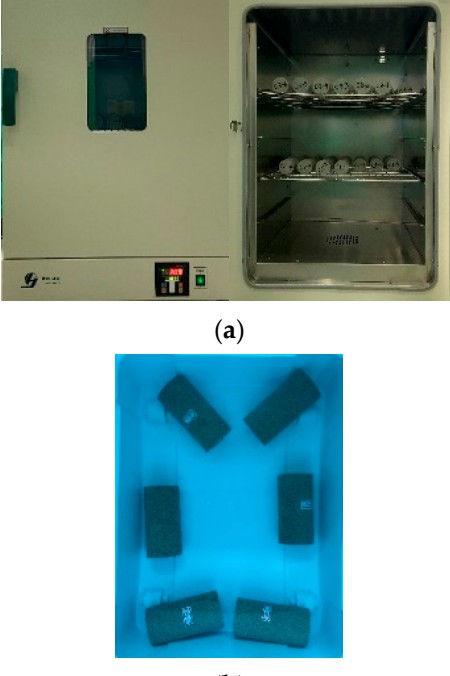

**Figure 2.** Specimen drying and immersion. (**a**) Drying. (**b**) Immersion.

### 2.2. Mechanical Tests

The RMT-301 mechanical test system (Figure 3) located at the national lab was adopted to conduct the mechanical tests.

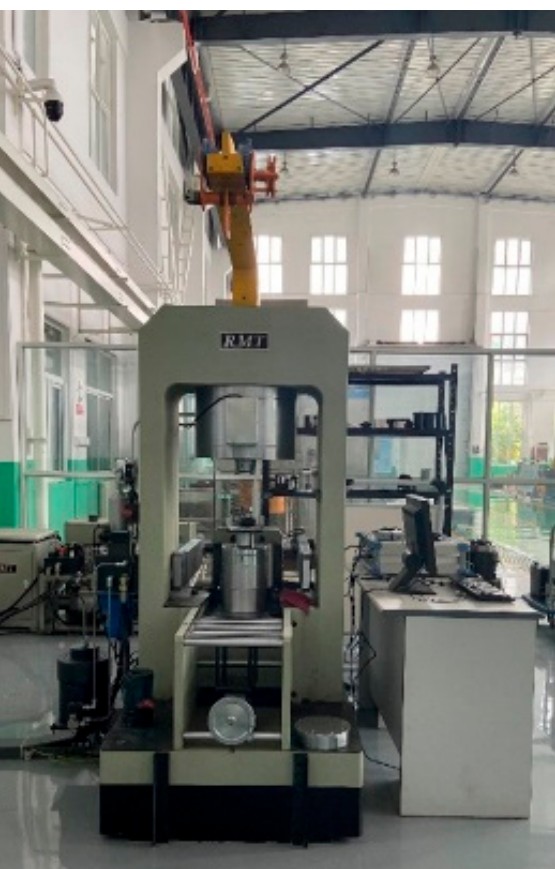

**Figure 3.** RMT-301 rock and concrete mechanics experimental system.

The uniaxial compression test: Under the displacement control model, the load was increased at the rate of 0.5 mm/min axially until the failure of the specimen.

The conventional triaxial compression test: (1) Under the stress control model, the confining pressure ($\sigma_2 = \sigma_3$) and the axial pressure were increased at the rate of 0.05 MPa/s alternatively at an interval of 5 MPa until the refining pressure reached the desired value (10 MPa, 20 MPa, and 30 MPa). (2) Under the displacement control model, while the refining pressure was maintained, the load was increased at the rate of 0.5 mm/min axially until the specimen failed.

The constant axial pressure and unloading confining pressure test: (1) Under the stress control model, the confining pressure $\sigma_2 = \sigma_3$ was increased at the rate of 0.2 MPa/s until the refining pressure reached the 20 MPa. (2) While the refining pressure was maintained, the axial load $\sigma_1$ was increased at the rate of 0.2 KN/s until the specimen reached 80% of the peak strength. (3) Under the displacement control model, while the axial pressure was maintained, the refining pressure was decreased at the rate of $v_3$ (0.005 MPa/s, 0.05 MPa/s, and 0.5 MPa/s until the failure of the specimen.

## 3. The Mechanical Analysis of Cementitious Sandstone

### 3.1. Mechanical Property Analysis

Based on the test data, the stress-strain curves were developed under various stress paths, as shown in Figure 4.

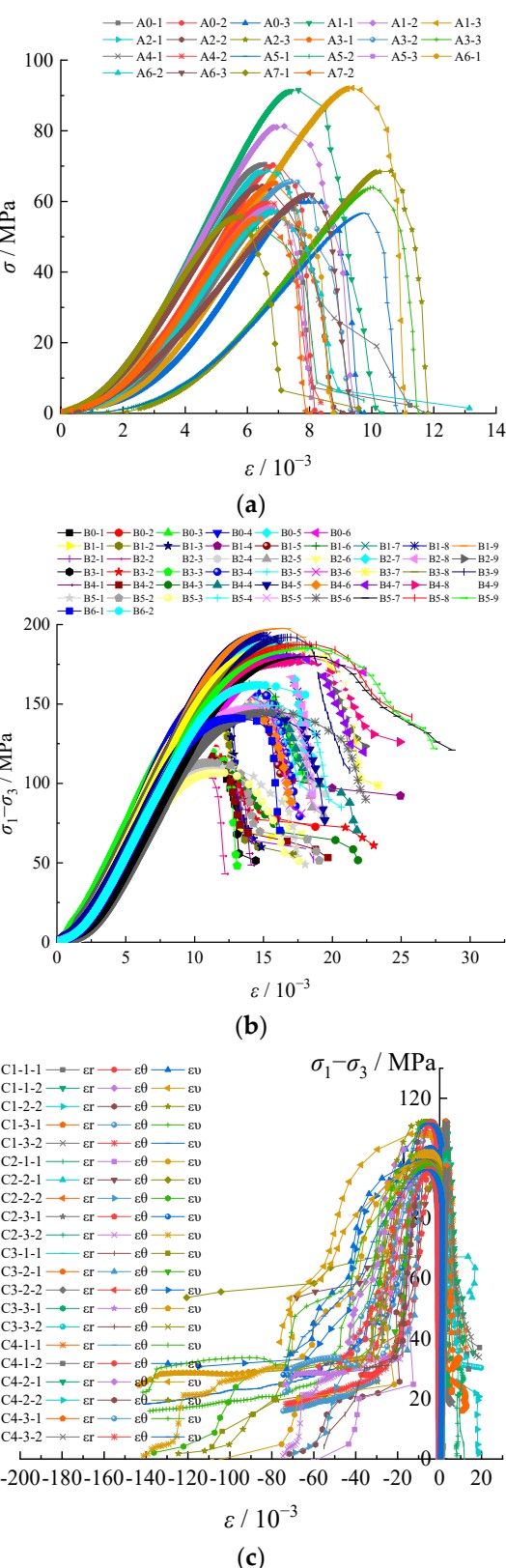

**Figure 4.** Stress-strain curves of rock specimens with different water content in uniaxial compression tests. (**a**) Uniaxial compression test. (**b**) The conventional triaxial compression test. (**c**) A constant axial pressure and an unloading confining pressure.

In order to further analyze the impacts of water content on the mechanical properties of cementitious sandstone, the relationships between the failure strength, elastic modu-

lus, Poisson's ratio, and water content of the specimens under different conditions were calculated and plotted, as shown in Figure 5.

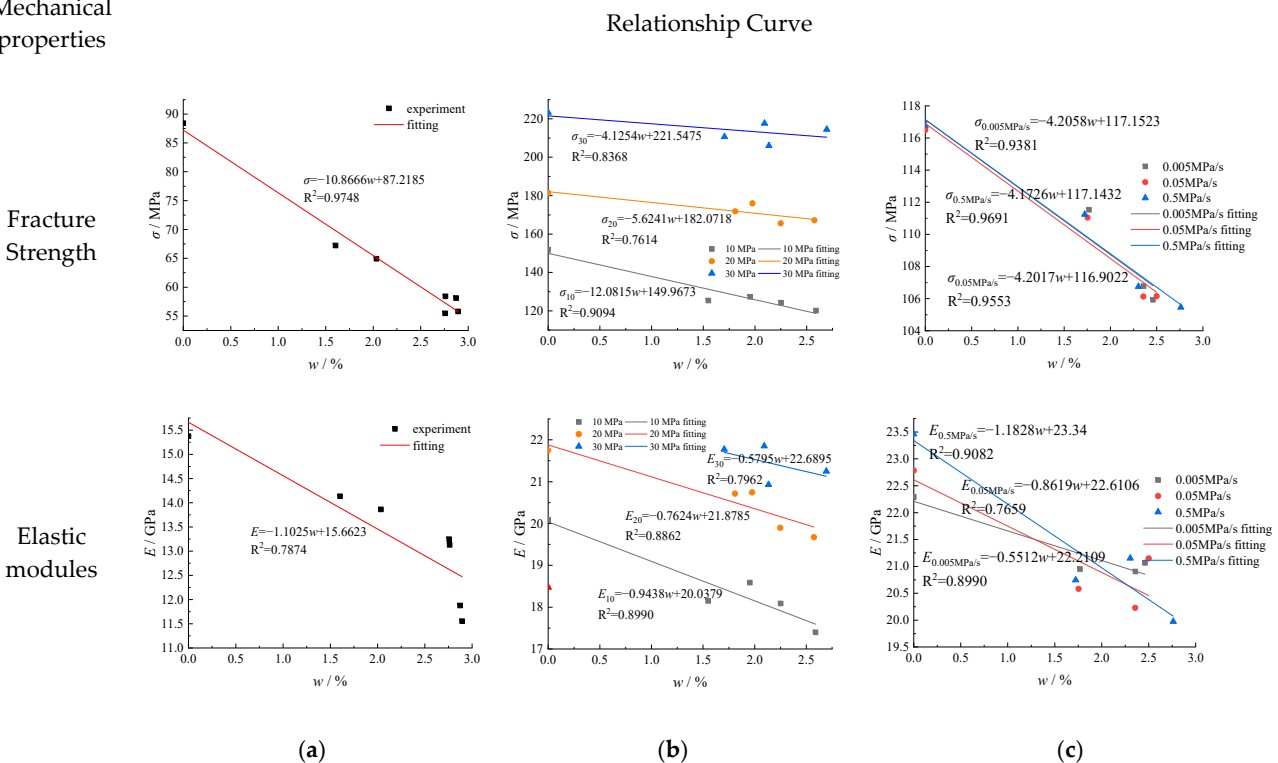

**Figure 5.** Relationship between mechanical parameters and water content. (**a**) Uniaxial compression test. (**b**) Triaxial compression test. (**c**) Const axial pressure and unloading confining pressure.

As shown in Figure 5, with the increase of water content, despite various stress path conditions, both the strengths and elastic modulus of rock mass specimens demonstrate a downward trend, with a strong linear correlation, indicating that water had a significant weakening effect on rock mass. From Figure 5b, the linear weakening relationship is better under the condition of low confining pressure. Meanwhile, the increase of confining pressure tends to inhibit the weakening effect of water on the rock mass. Therefore, in spite of the water content, the increasing of the refining pressure produces a restraining effect on the rock damage. The combination of Figures 4c and 5c reveals that the lateral strain of the specimen increases rapidly during the confining pressure unloading, indicating some significant capacity expansion characteristics. This further shows that under the combined action of the water environment and confining pressure unloading, the rock mass tends to reach the failure limit at a faster pace, accompanied with a strong weakening effect.

### 3.2. The Study of Strain Softening Constitutive Mode

According to the mechanical tests of cementitious sandstone, obvious turning points in the mechanical property in the stress-strain curves were identified. As demonstrated in Figure 6, taking the dry specimens as an example, five basic stages including the compaction stage, the linear elastic stage, the strain hardening stage, the strain-softening stage, and the residual stage can be identified.

$$
\begin{cases}
\sigma = E_{AB}\varepsilon, 0 < \varepsilon \leq \varepsilon_e \\
\sigma = \sigma_e + E_{BC}(\varepsilon - \varepsilon_e), \varepsilon_e < \varepsilon \leq \varepsilon_c \\
\sigma = \sigma_c + E_{CD}(\varepsilon - \varepsilon_c), \varepsilon_c < \varepsilon \leq \varepsilon_r \\
\sigma = \sigma_r + E_{DE}(\varepsilon - \varepsilon_r), \varepsilon > \varepsilon_r
\end{cases}
\tag{1}
$$

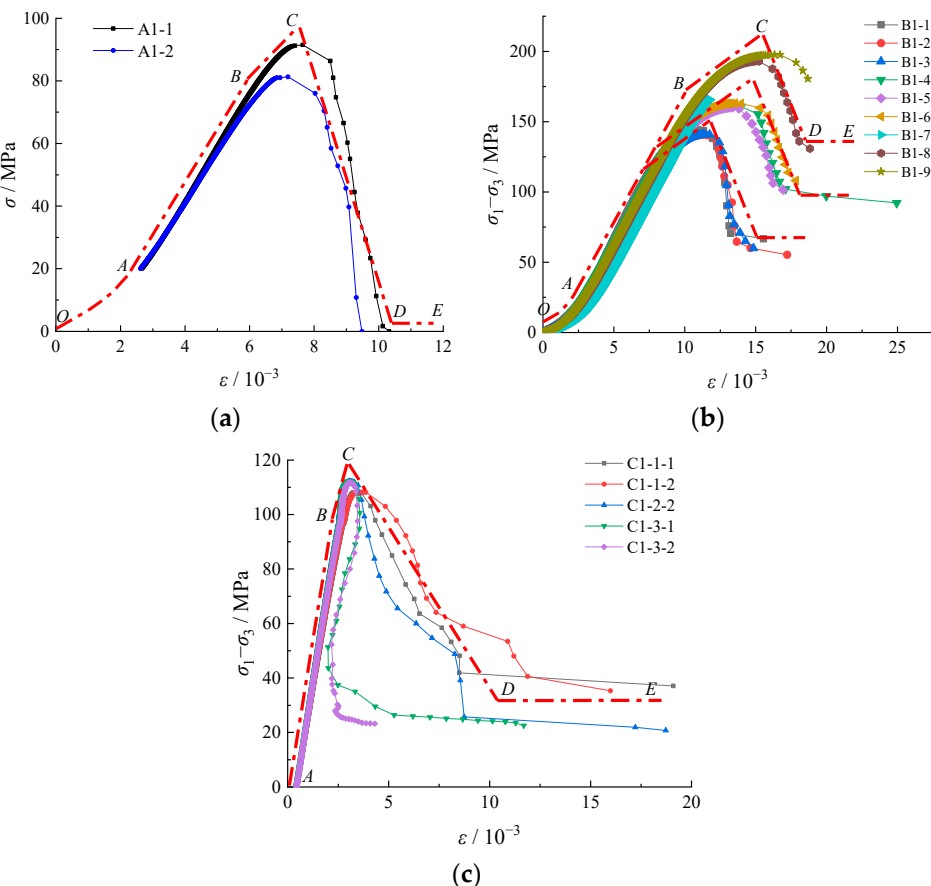

**Figure 6.** Stress-strain curve and mechanical turning point. Since the specimens were compacted before the application of the confining pressure and the hydraulic pressure in the conventional triaxial compression test, the compaction stage was excluded from this study. A multi-linear, strain-softening model of cementitious sandstone was established. (**a**) Uniaxial compression. (**b**) Conventional triaxial compression. (**c**) A constant axial pressure and an unloading confining pressure.

In the equation, $\varepsilon_e, \varepsilon_c, \varepsilon_r$ correspond to the strain at point B, C, D in the stress-strain curve. $\sigma_e, \sigma_c, \sigma_r$ refer to the stress corresponding to the strain respectively. $E_{AB}, E_{BC}, E_{CD}, E_{DE}$ refer to the variation rate of the stress corresponding to the strain in the linear elastic stage AB, strain hardening stage BC, strain softening stage CD and residual stage DE in the stress-strain curve, respectively.

In order to analyze the strain variation rate corresponding to the stress, Equation (1) can be modified into:

$$\begin{cases} E_{AB} = \frac{\Delta\overline{\sigma_{AB}}}{\Delta\varepsilon}, 0 < \varepsilon \leq \varepsilon_e \\ E_{BC} = \frac{\Delta\overline{\sigma_{BC}}}{\Delta\varepsilon}, \varepsilon_e < \varepsilon \leq \varepsilon_c \\ E_{CD} = \frac{\Delta\overline{\sigma_{CD}}}{\Delta\varepsilon}, \varepsilon_c < \varepsilon \leq \varepsilon_r \\ E_{DE} = \frac{\Delta\overline{\sigma_{DE}}}{\Delta\varepsilon}, \varepsilon > \varepsilon_r \end{cases} \qquad (2)$$

In the equation, $\overline{\sigma}$ refers to the main stress difference, which is $\overline{\sigma} = \sigma_1 - \sigma_3$. $\Delta\overline{\sigma_{AB}}, \Delta\overline{\sigma_{BC}}, \Delta\overline{\sigma_{CD}}, \Delta\overline{\sigma_{DE}}$ refer to the main stress difference in the linear elastic stage AB, strain hardening stage BC, strain-softening stage CD, and residual stage DE in the stress-strain curve, respectively. $\Delta\varepsilon$ refers to the strain difference corresponding to the $\Delta\overline{\sigma}$.

According to the test data, the relationships between the specimen strength, the variation rate, and the water content under various stress paths were developed and presented in Figures 7–9 below.

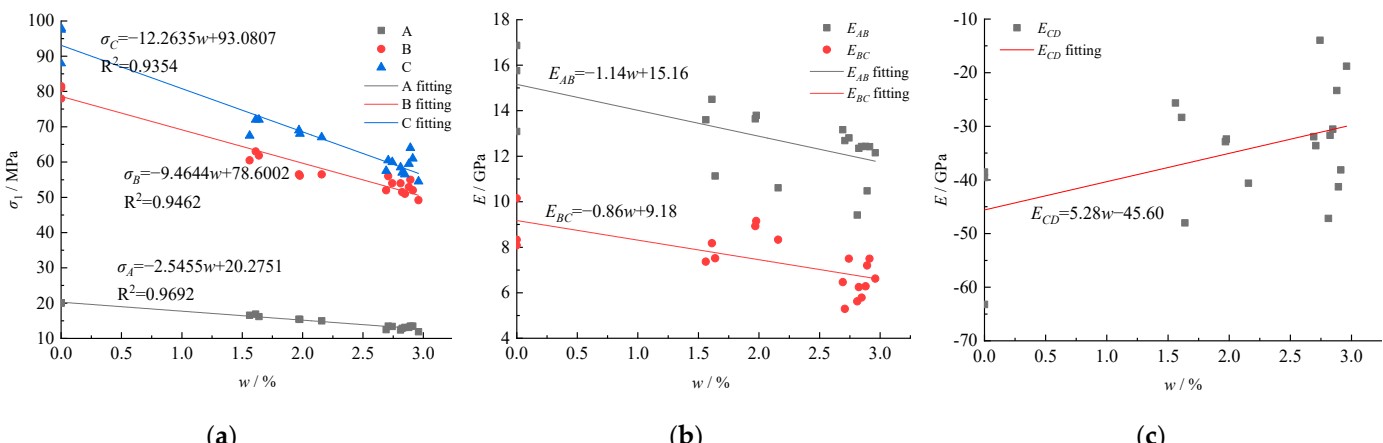

**Figure 7.** The mechanical turning points of the specimen under uniaxial compression, and the strength variance rate corresponding to the water content. (**a**) Mechanical turning point. (**b**) Stage AB and BC. (**c**) Stage CD.

**Figure 8.** The mechanical turning points of the specimen under triaxial compression, and the strength variance rate corresponding to the water content. (**a**) Strength at the mechanical turning point. (**b**) Strength variance rate.

Unloading
rate/(MPa/s)

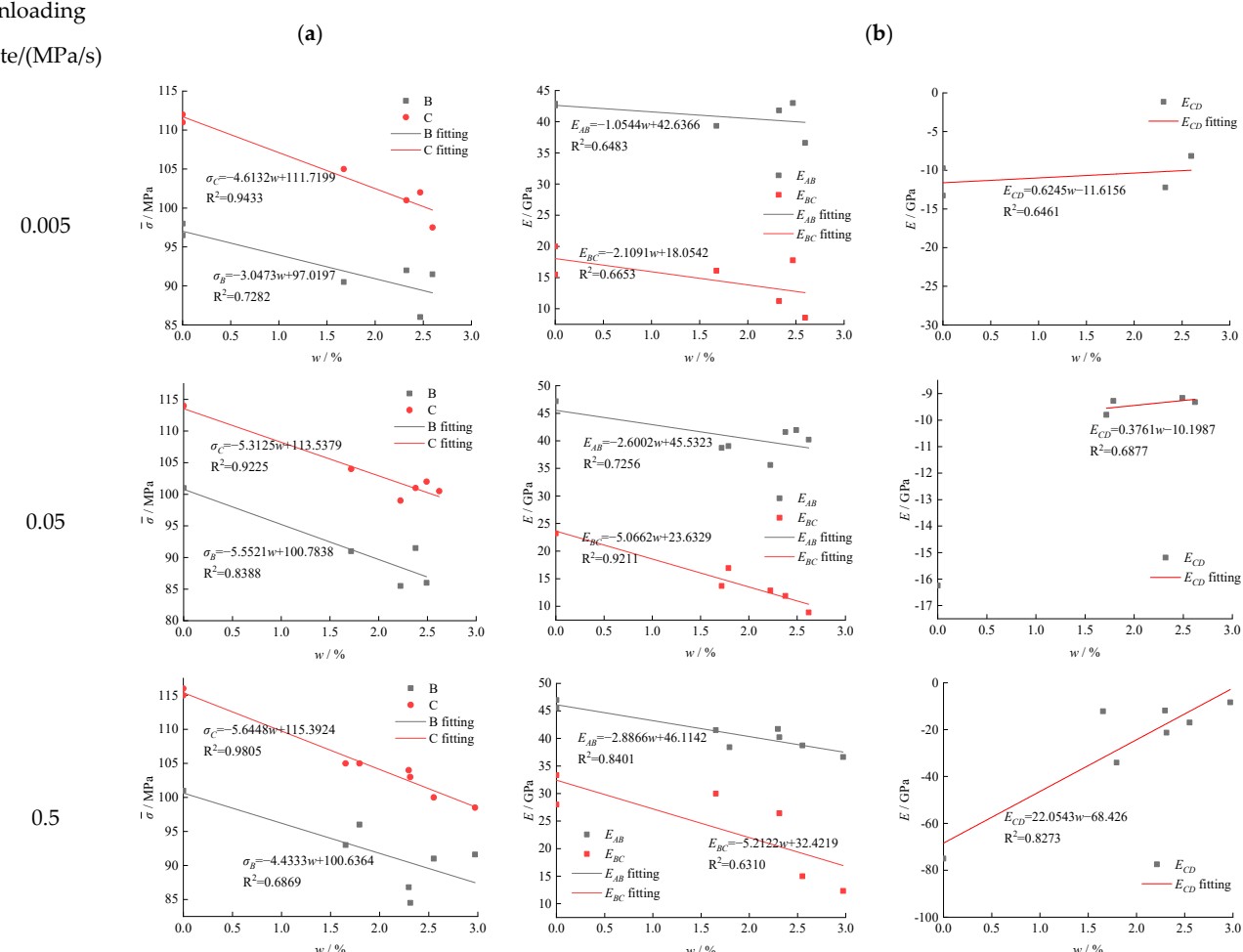

0.005

0.05

0.5

**Figure 9.** The mechanical turning points of the specimen under constant axial pressure, an unloading confining pressure, and the strength variance rate corresponding to the water content. (**a**) Mechanical turning point. (**b**) Strength variance rate.

According to Figure 7a, with the increase of water content, the turning points of the rock mass show a consistent linear decreasing relationship at each stage, indicating that the water has a weakening effect on the rock mass, which was observed at any moment when the rock mass was stressed. In terms of the water weakening effect, the weakening effect reaches the peak at failure point C, followed by the elastic-plastic turning point B and the compression elastic turning point A. From Figure 7b,c the strength variance rate of the specimen at each stage also decreases with the increase of water content, and the slopes reflect the speeds of strength reduction and weakening. Consistently with 7a, the weakening effect tends to grow stronger with the increase of the water content. Meanwhile, from Figure 7c, the hydration transforms the specimen from brittle failure to ductile failure.

According to Figure 8a, under various confining pressure conditions, the mechanical turning points of the cementitious sandstone are consistent with changes in the water content. The mechanical strength tends to decrease along with the increase of the water content. As shown in Figure 8a, point A (condensation-linear elastic turning point) and point C (peak strength point) demonstrated a high correlation with water content, and with the fitting coefficients above 0.85. At point B, the fitting coefficient corresponding to the water content variance is relatively lower, indicating a bigger role played by the environment and the specimen itself. According to Figure 8b, under one confining pressure, with the increasing of the water content, the variance rate of the specimen decrease, indicating a limited water weakening effect and the existence of the water saturation point. When the water content in the specimen exceeds the threshold value, the water no longer impacts

the mechanical properties. The confining pressure tends to inhibit the weakening effect of the water on the specimen. Under similar or the same water content, a higher confining pressure slows down the decrease of the mechanical strength, which is consistent with the enhancing effect of the confining pressure.

According to Figure 9a, under a constant axial pressure combined with an unloading confining pressure, along with various unloading rates, the mechanical turning point of the cementitious sandstone specimen demonstrates a consistent linear decreasing relationship with the water content. A longer soaking leads to lower stress at the mechanical turning point at various stages. When the rock reached saturation status, no obvious change was observed in the mechanical properties. At this point, the rock mass becomes the only relevant factor. From Figure 9b, under various unloading rates, a linear decreasing relationship is identified between the specimen strength variance rate and the water content. In other words, as the water content increases, the stresses in various stages tend to decrease until the saturation point was achieved.

In summary, the established, linear elastic-strain, hardening-strain, softening-residue, multi-linear, constitutive equation has good adaptability to stress-strain curves of cementitious sandstone under uniaxial, triaxial, and unloading conditions. At the same time, the stress value of the turning point in the multi-linear strain-softening model can also characterize the relationship between water content and stress path on the mechanical properties of rock.

### 3.3. A Discussion of the Failure Mode

From a microscopic point of view, the damage of rock refers to the generation and expansion of microcracks. From a macroscopic point of view, the damage of rock can be viewed as the continuous expansion of microcracks into a visible rupture surface. The failure mode of rock can be reflected by the macroscopic fracture surface formed after the rock mass specimen loses its bearing capacity. The failure states of the specimens under different stress path conditions are shown in Figure 10.

As demonstrated in Figure 10a, the majority of the failures of rock mass specimens under uniaxial compression are splitting failures, which are nearly parallel to the axial direction, with some shear cracks dominated by the tensile failure. The damage occurred due to the fact that the tensile strength of the rock is much smaller than the compressive strength. Another explanation lies in the decrease in the uniaxial compressive strength of the specimen, caused by the internal tensile failure in the specimen. According to Figure 10b, under the 20 MPa confining pressure, the dry specimen exhibits conjugate shear failure accompanied with brittle splitting failure on the surface. When the specimen was immersed in water for 0.75 h, the rupture surface of the specimen was featured as shear failure. When the immersion time was more than 1.5 h, the shear failure dominated the surface fracture. The fracture surface became falter, indicating that with the increasing water content, the failure mode of rock mass gradually transformed from tensile-shear composite failure to shear failure. According to Figure 10c, under a constant axial pressure and an unloading confining pressure, the dry specimen is dominated by shear failure. In other words, only one shear rupture surface is identified, accompanied by obvious debris peeling. When the specimen was immersed in water for 1.5 h, the tensile-shear composite failure occurred, accompanied by the conjugate shear failure and local tensile failure. At this time, the failure forms were considered more discrete. When the specimen was immersed in water for 6 h, the shear failure became obvious again, accompanied by local tension cracks. When the specimen was immersed in water for 24 h, the specimen was mainly damaged by shear tension, with not only axial tension cracks but also circumferential tension cracks locally. Expansion was observed in the middle. Meanwhile, in the unloading test, the corresponding relationship between the fracture angle, the water content, and the unloading rate of the confining pressure was not obvious. As the unloading rate of the confining pressure increased, the shear damage become obvious along with smoother fractures. Combined with the multi-linear strain-softening model, it is concluded that under

different stress paths, the increase of water content weakens the brittle failure characteristics of sandstone.

In addition, under unloading conditions, the failure modes of rock mass specimens were dominated by brittle failures which were a mix of tensile, shear, and tensile-shear composite fractures. During the unloading process, progressive failures were observed. Generally, a clear sound caused by the release of crack propagation energy can be heard before failure. During the failure, the bearing capacity of the rock sample decreases rapidly or even is lost completely, resulting in brittle fracture and a crisp cracking sound. The dynamic characteristics displayed are caused by the sudden release of the energy accumulated during the loading process. Compared with loading failure, unloading failure is more sudden and drastic.

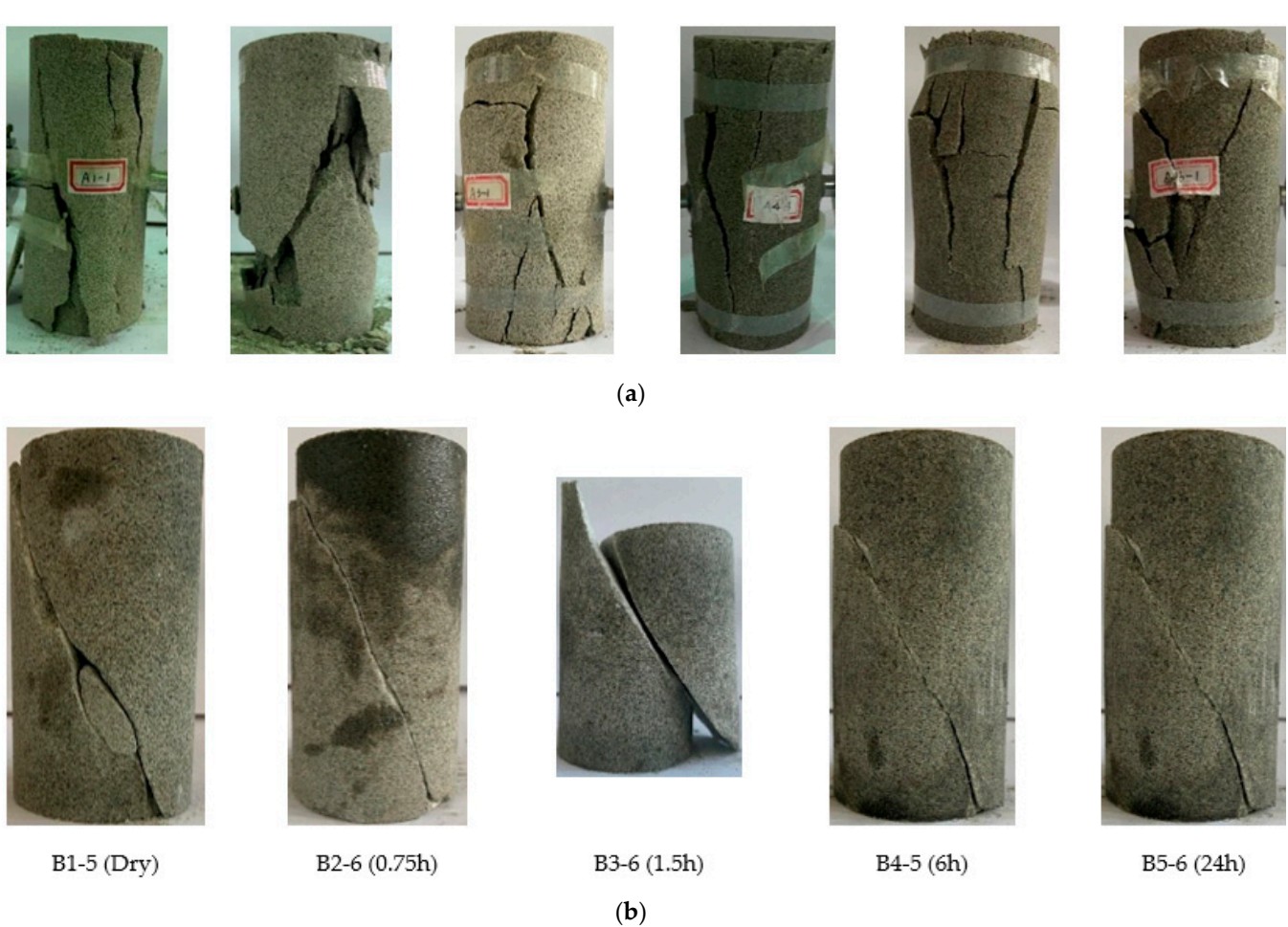

(a)

B1-5 (Dry)     B2-6 (0.75h)     B3-6 (1.5h)     B4-5 (6h)     B5-6 (24h)

(b)

**Figure 10.** *Cont.*

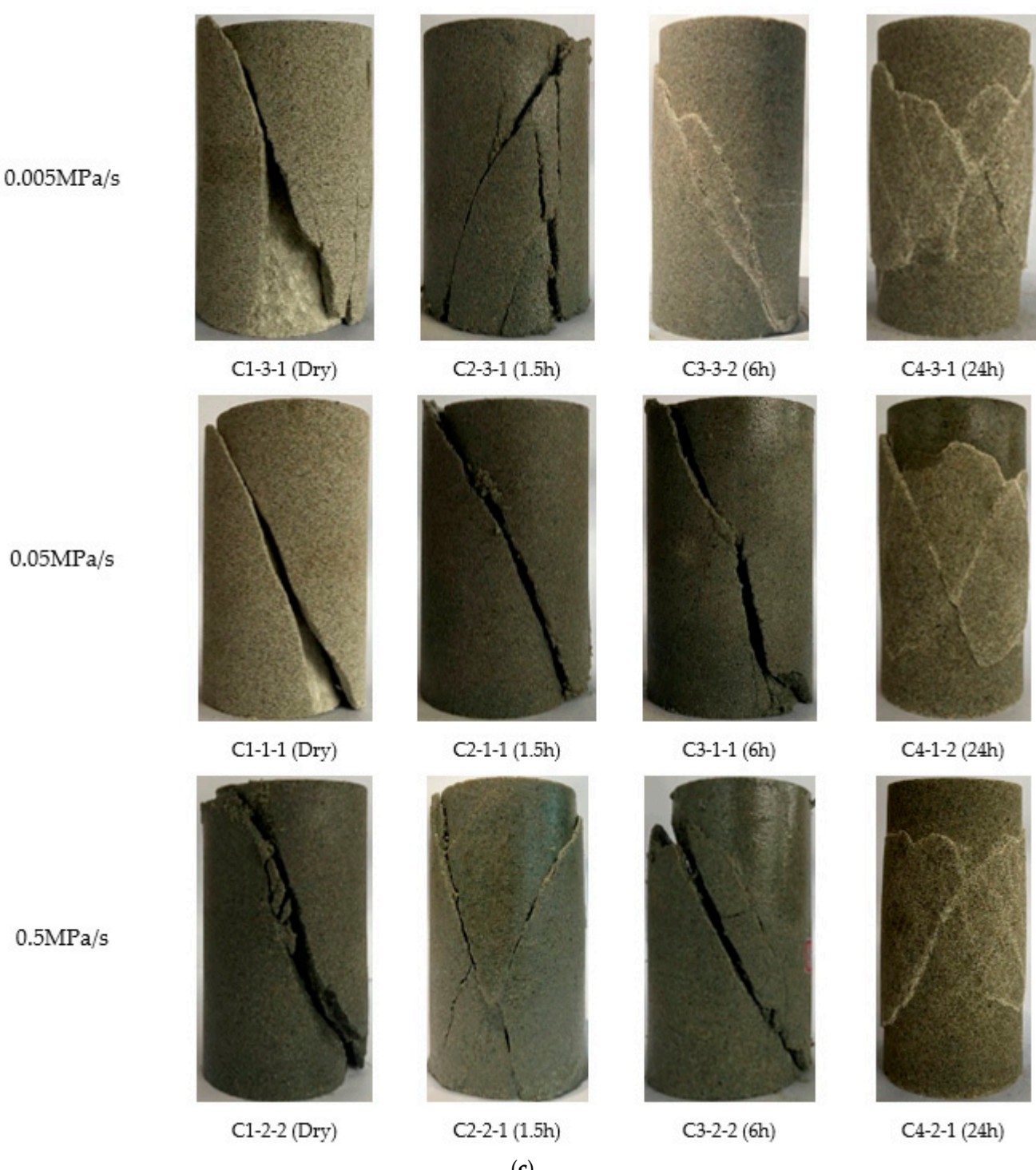

**Figure 10.** Relationship between mechanical parameters and water content. (**a**) The uniaxial compression test. (**b**) The conventional triaxial compression test. (**c**) A test of constant axial pressure and unloading confining pressure.

## 4. Conclusions

Based on the conventional triaxial compression test and the constant axial pressure test on the cementitious sandstone, along with the establishment of a multi-linear, strain-softening constitutive model, the following conclusions are drawn:

1.  With the increase of the water content, the rock strength and elastic module tend to decrease linearly, demonstrating a strong water weakening effect. The surrounding

rock produces an inhibiting effect on the rock failure while the unloading process of the confining pressure expedites the water weakening process to the rock mass.

2.  A multi-linear, strain-softening constitutive model was established based on the stress-strain curves. Under various stress paths, the turning point strengths in various stages of the rock and the strength variance rates tend to decrease linearly along with the increase of the water content. This indicates the weakening effect of the water content on the rock, especially the fracture points of the rock mass. Due to the water content, the failure mode of rock mass gradually transformed from brittle failure to ductile failure.

3.  Under the conventional triaxial compression, cementitious sandstone is dominated by tensile fractures. Under the conventional triaxial compression, the failure mode of rock mass gradually transforms from the tensile-shear composite failure to the shear failure. Under a constant axial pressure and an unloading confining pressure, the expansion failure dominates, along with dynamic failure characteristics. However, the increase of water content weakens the brittle failure characteristics of rock mass under different stress paths.

**Author Contributions:** Methodology, W.H.; formal analysis, Y.K.; writing—original draft preparation, Z.S.; writing—review and editing, K.W. All authors have read and agreed to the published version of the manuscript.

**Funding:** This research was funded by [the Science innovation and entrepreneurship special funded projects of China Coal Technology& Engineering Group] grant number [2020-TD-ZD007], [the Natural Science Foundation of Chongqing, China] grant number [cstc2020jcyj-msxmX0972, cstc2020jcyj-msxmX0676 & cstc2019jcyj-msxmX0633], and [the Technology Innovation and Application Development Project of Chongqing, China] grant number [cstc2020jscx-msxmX0216].

**Data Availability Statement:** The data used to support the findings of the study are available from the corresponding author upon request.

**Conflicts of Interest:** The authors declare that no conflict regarding the publication of this paper has been identified.

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
