# Peer review of "A Study on the Strain-Softening Constitutive Model of Cementitious Sandstone"

_water, doi:10.3390/w14081309_

Round 1

Reviewer 1 Report

Comments are attached

Author Response

Dear Reviewers,

Firstly, please allow me to thank you for your review and detailed feedbacks.

Secondly, based on your feedbacks, the following modifications have been made to the draft.

(1) The innovative highlights of this study centers around the Strain Softening Constitutive Model. With this model, the mechanical turning points demonstrated in the test samples under various stress path conditions along with the variance rates corresponding to different water contents indicate a consistent weakening effect of water on the rocks.

(2) The abstract and introduction were reorganized and updated.

(3) The charts and figures were updated in English.

(4) Some modifications have been made to the language of the draft.

Thank you again.

All authors as follows: Zhongguang Sun, Kequan Wang,

Yueming Kang and Wanli Hu

Reviewer 2 Report

The paper presents a study based on triaxial tests carried out on a particular sandstone. The setting of the tests is quite appropriate, but the introduction must be improved in order to better contextualise the research, which in this way appears just theoretical. At present stage the part regarding the failures and underground construction implications is not consistent.

General comments

- Line numbers are missing, making difficult the revision process

- The graphs in the figures starting from 4 contain Chinese characters, this should be fixed

Specific comments

- I don’t understand the need of the first sentence with the topic of the paper. Moreover, the citation

related to that appears not appropriate (probably citation 1 should be 2 and viceversa)

- Authors should make a clear distinction between rock and rock mass. Then they should explain the sentence about adding complexity to underground engineering, as it is not very clear from the text

- The introduction describes the influence of the water on the strength of the rock, by stating also that 80% of rock slope failures are due to this. But also in this case most of the failures are due to instabilities along joints, thus an explanation should be given. Then considering the intact rock, some concerns about the porosity should be introduced here, and the fact that the water influences only certain type of intact rocks (like sedimentary)

- in Figure 5 the first line is not clear (mechanical properties/relationship curve

Author Response

Dear Reviewers

Firstly, please allow me to thank you for your review and detailed feedbacks.

Secondly, based on your feedbacks, the following modifications have been made to the draft.

(1) The charts and figures have been updated in terms of language.

(2) The first a few sentences included in the introduction mean to indicate a strong connection between the successes of the underground projects and the mechanicals of rocks, which has been modified and updated. Reference 1 and 2 were double checked and verified. 

(3) The failure of the slope is mainly caused by the instability of the rock itself. However, with the water, the failure process is expediated, indicating that the water has produced a weakening effect to the rock strength.  

(4) In figure 5, the mechanical parameters refer to the fracture strength and elastic modulus. The relationship curves mean to demonstrate the relationship between the stress indicators and the water contents under three stress paths.

Thank you very much again.

All authors as follows: Zhongguang Sun, Kequan Wang,

Yueming Kang and Wanli Hu

Reviewer 3 Report

Authors should justify why they chose such rocks for research. Is there going to be an investment in the mine, for which research should be done? Indicate from which seam and depth the samples were taken. Indicate the location of the samples. Justify that they are representative of the geological layer. Eliminate Chinese language markings from diagrams. 

Author Response

Dear editors,

Firstly, please allow me to thank you for your review and detailed feedbacks.

Secondly, based on your feedbacks, the following modifications have been made to the draft.

(1) Additional information regarding the rock specimen selection and preparation has been added to the draft.

(2) The charts and figures have been updated in terms of language.

Once again, thank you for your patience and help.

All authors as follows: Zhongguang Sun, Kequan Wang,

Yueming Kang and Wanli Hu

Round 2

Reviewer 1 Report

Thank you very much for updating the manuscript. I have the following comments on the revised version;

  1. I can’t see the responses explicitly.
  2. The pdf document shows only English language modification.
  3. The abstract and the manuscript still missing the research gap. As the aim of the study is to study insight, therefore the document is more like a technical report, rather than an innovation.
  4. Mention briefly the significance and the application of the research.
  5. Discuss the results with the innovations.

Author Response

Dear Reviewer,

Thank you for your comments again, I have made revisions as detailed in the manuscript.

Since some of the previous comments have different opinions from those of other reviewers, a compromise was made during the revision, so it seems not obvious to modify. But I still very grateful for your contribution to the manuscript. I also hope in the later exploration and research process can get your help and guidance.

All authors as follows: Zhongguang Sun, Kequan Wang, Yueming Kang and Wanli Hu

Reviewer 2 Report

Thank you for the updated version and the replies. I'm not 100% agreeing on the statement (3), as I think that the water along the joints is playing a major role on the slope stability. Anyway I'm satisfied with the improvement

Author Response

Dear Reviewer,

Thank you for your contribution to the manuscript.

Regarding statement (3), I will discuss it more deeply in the later research, and I also hope to get more advice and guidance from you, thank you again.

All authors as follows: Zhongguang Sun, Kequan Wang, Yueming Kang and Wanli Hu
